# Mechanical Properties and Fracture Microstructure of Polycarbonate under High Strain Rate Tension

**DOI:** 10.3390/ma16093386

**Published:** 2023-04-26

**Authors:** Sai Zhang, Bingqian Wang, Xianming Meng, Yajun Chen

**Affiliations:** 1China Automotive Technology and Research Center Co., Ltd., Tianjin 300300, China; zhangsai@catarc.ac.cn; 2Sino-European Institute of Aviation Engineering, Civil Aviation University of China, Tianjin 300300, China

**Keywords:** polycarbonate, composite material, high-speed tensile testing

## Abstract

In this paper, static and dynamic tensile tests were conducted on two kinds of polycarbonate (HL6157 and A1225BK), combined with the digital image correlation (DIC), for guiding the development of the battery pack of new energy vehicles. The mechanical properties of polycarbonate at low-speed (0.01/s) and high-speed (1/s, 100/s) tension were investigated and the microstructure of the fracture for polycarbonate at different speed tensions was also investigated. The fracture microstructure of two kinds of materials was also investigated in this paper. The tension results showed that as the strain rate increased, the yield strength and modulus increased, and the yield strength of the two materials increased by 30% under high-speed tension. In addition, the fracture strain increase was greater than 10% as the strain rate increased. Meanwhile, for polycarbonate, the strain rate increased, and the fracture toughness increased.

## 1. Introduction

At present, new energy vehicles and other new energy projects are developing rapidly. At the same time, the weight of new energy vehicles will affect the endurance of new energy vehicles. Therefore, the lightweight design of vehicles is the focus of current research. In addition, lightweight new energy vehicles need to consider the safety problems during driving, such as the stone impact of the battery pack; it is the key to the current research to find new materials to reduce the overall weight of new energy vehicles while ensuring safety during driving [1,2]. Composite materials have excellent specific strength and stiffness, which can achieve the goal of reducing the overall weight of the structure [3,4,5]. As an important class of composite materials, carbon fiber composite materials can be applied to the battery pack shell to meet the weight reduction requirements of new energy vehicles, but the impact resistance of traditional carbon fiber composite materials is poor [6,7], and carbon fiber is a brittle material, so in order to improve the impact resistance of carbon fiber composite materials, resin with better impact resistance is the key research object. In the impact process, such as stone impact, the strain rate is high, and the mechanical response, failure mode, and static state of the material are completely different under high strain rate. In order to be closer to the actual impact condition, it is worthwhile to investigate the performance of resin under high strain rates. Therefore, a high-speed tensile test was used to study the performance of the resin at a high strain rate.

Polycarbonate (PC material) is a typical thermoplastic resin. Compared with thermosetting resins, thermoplastic resins have the following remarkable characteristics in terms of mechanical properties: ① they have obvious mechanical relaxation; ② elongation at break; and ③ good impact resistance. As such, PC material can be used as a composite material matrix to improve the impact resistance of the material. At present, there has been a large number of studies on the high-speed tensile properties of PC materials and thermoplastic resins [8,9,10]. Sarva et al. [11] used a separate Hopkinson rod to conduct dynamic tensile experiments on PC materials at room temperature and high strain rates and proposed that PC materials are pressure sensitive. Wang et al. [12] studied the compressive properties of polycarbonate at different strain rates and proposed a ZWT constitutive model for describing the compression of medium strain rates. Li et al. [13] used the improved split Hopkinson rod test combined with digital image correlation (DIC) to study the tensile properties of polycarbonate. Zhu et al. [14] studied the high-speed tensile properties of traditional thermoplastic resin polypropylene, and believed that the tensile strength of polypropylene resin increased linearly with the logarithm of the strain rate. Yu et al. [15] summarized the high-speed test methods and corresponding damage forms of polycarbonate. Liu et al. [16] studied the SHPB technique for the measurement of the dynamic strength of polymers, and used an iterative correction methodology to identify the actual strain-rate effect. Yang et al. [17] studied the relationship between temperature and strain rate and gave the expression which can be used to describe the relationship between yield strength and strain rate. At present, although people have fully studied the mechanical properties of PC-based amorphous polymer materials, the data from tensile experiments at high strain rates are still relatively scarce, which limits the wide application of PC-based resins. The dynamic mechanical properties are still lacking comprehensive and systematic research.

In order to guide the follow-up lightweight design of new energy vehicles and build a battery pack with composite materials using PC materials as the matrix, this paper used a high-speed tensile testing machine and two candidate polycarbonate resins and conducted tests at low and high speeds. In addition, digital image correlation (DIC) was used to capture the strain field for analyzing the difference of the strain distribution at different strain rates. The comparison of the tensile properties of the PC material, the tensile failure behavior, and the change of the mechanical properties of the PC material at high strain rates was studied.

## 2. Materials and Methods

### 2.1. Materials

Polycarbonate has good impact resistance, heat distortion resistance, good weather resistance, and high hardness, so it is suitable for the production of various parts of cars and light trucks, mainly in lighting systems, instrument panels, heating panels, and polycarbonate alloy bumpers, etc. This paper mainly studied the mechanical properties of two PC materials, HL6157 and A1225BK, under different strain rate conditions.

HL6157 material has the following advantages: (1) Extremely low impurity content: the unique reaction and devolatilization technology can significantly reduce the impurity content, increase PC material transmittance (≥90% ASTM D1003), and reduce haze (≤0.5% ASTM D1003). (2) Excellent aging resistance: unique high end-capping rate and special additive system, with excellent UV resistance and heat and humidity resistance, and long-term use without performance degradation and other problems. (3) Excellent mechanical and heat resistance properties: notched impact strength up to 850 J/m (ASTM D256), and heat distortion temperature ≥ 130 °C (ASTM D648).

A1225BK material has the following advantages: (1) High impact: the notched impact strength can reach 750 J/m (ASTM D256), and it has excellent falling ball impact strength. (2) High gloss (gloss ≥ 105GU ASTM D523) and no pitting: professional dyeing technology and high-quality mixing production line are adopted, the colorant is evenly distributed, and the brightness is higher; the product has no pitting and defects are not visible to the naked eye. (3) High fluidity: high-flow PC material is adopted, with narrow molecular weight distribution and excellent product fluidity. (4) High heat resistance: the heat resistance is 3–5 °C higher than that of the same grade of trans-esterified PC, and the heat distortion temperature is ≥128 °C (ASTM D648), which meets the requirements of use.

Both materials have high impact resistance and can be used as candidate materials in an automotive lightweight design. The basic information and process parameters are shown in Table 1.

### 2.2. Methods

The high-speed tensile test of the resin refers to the standard WSS-M99D68-A1. The geometric dimensions of the sample are shown in Figure 1; the thickness is 3.0 mm, and the tensile strain rates of the specimen are 0.01/s, 1/s, and 100/s. Five specimens were tested under each tensile condition, and the average was selected as the final result.

The quasi-static tensile test was carried out on the SANSI universal material testing machine (SANSI, Shenzhen, China). The loading frame of the SANSI electronic universal testing machine was jointly controlled by both sides, so that the load could be loaded relatively smoothly, and it also ensured that the sample had a good simultaneous loading process. In addition, high-speed tension was carried out under the high-speed tension machine Zwick-HTM-16020 (ZwickRoell, Ulm, Germany). The upper fixture of the high-speed tension machine was a squirrel cage fixture for high-speed displacement loading, and the lower fixture was a load sensor. In order to capture the load changes during the test, the cold light lamps on both sides provided the light source for the DIC shooting.

The strain rate in this paper is calculated according to Equation (1),
(1)$$\dot{\varepsilon }=\frac{\Delta \varepsilon }{\Delta t}=\frac{\Delta l}{l_{0}}\times \frac{1}{\Delta t}=\frac{\Delta l}{\Delta t}\times \frac{1}{l_{0}}=\frac{v}{l_{0}}\;$$
where $\dot{\varepsilon }$ is strain rate, $s^{-1}$; Δε is strain increment; Δt is time increment, s; Δl is the deformation increment in Δt, mm; $l_{0}$ is initial length of specimen, mm; *v* is the loading speed, mm/s.

Due to the high-speed tensile test, it is difficult to capture the effective strain, so DIC technology is used to collect the s strain during the tension. The DIC-based high-speed tensile test platform is shown in Figure 2. We sprayed white paint and black spots on the front and back of the sample. We used a DIC device camera (ZwickRoell, Ulm, Germany) to collect scattering image information fixed on the side of a high-speed tensile testing machine. We used the strain cloud map and obtained the strain during the tension.

Gold spraying was applied to the fracture surface of the specimen at low and high speed; the Hitachi S-3400N scanning electron microscope (Hitachi, Tokyo, Japan) was used for micro-appearance characterization. The fracture morphology of resins at various scales under different material types and different strain rates was observed, and we analyzed the effects of different materials and different strain rates on the tensile fracture of resins.

## 3. Results and Discussion

### 3.1. Stress-Strain Curves of Resins at Different Materials and Strain Rates

Figure 3 depicts the stress–strain curve of PC material, which is a typical thermoplastic resin, divided into four stages: initial tension, softening after local yield, plastic strain expansion, and fracture. The global mechanical response was consistent with the results of Liu [16]. This phenomenon was more pronounced during static tension (0.01/s) and less observable at the final broken stage during high strain rates (1/s and 100/s). The possible reason is that the break mode of the resin changed due to different strain rates. Additionally, the yield strength of the resin significantly increased with the strain rate, while the yield strain exhibited a rising trend. The strain rate effect also affected the broken strain of the resin, with an approximately 10% increase observed for the A1225BK material and a 24% increase for the HL6157 material when shifting from static tension to dynamic tension. These results indicate that an increase in strain rate enhances the anti-fracture capacity of the material and that the A1225BK material possesses better anti-fracture capabilities than the HL6157 material, despite their similar mechanical properties.

### 3.2. Effect of Strain Rate on the Mechanical Performance

Since the performance of PC material changes as the strain rate increases, the overall trend is that as the strain rate increases, the mechanical properties of PC material such as tensile modulus and yield strength will increase. In order to study the change of elastic modulus and yield strength with strain rate, the logarithm of elastic modulus, yield strength, and strain rate were plotted, and the function fit was performed.

As shown in Figure 4, the logarithm of the elastic modulus of the PC material and the tensile strain rate was basically linear, and the basic form of the fitting formula is Equation (2), which is in consistent with the formula in Reference [17],
(2)$$y=a\times \log\dot{\varepsilon }+b$$
where a and b are obtained by numerical fitting. It can be obtained from the form of Equation (2), where b can reflect the tensile modulus of PC material at a strain rate of 1/s, and a can reflect the tensile modulus of PC material and the sensitivity to the strain rate, that the larger the value of a, the more sensitive the material is to the strain rate, and the tensile modulus changes more obviously with the strain rate. Therefore, it was concluded that the value of a for the A1225BK material was 174.95 and the value of a for the HL6157 material was 28.75, so the A1225BK material was more sensitive to the strain rate than the HL6157 material.

Similarly, the yield strength of the PC material was treated with the same as the tensile modulus, and the numerical fitting was carried out again. Results similar to the tensile modulus were obtained, as shown in Figure 5. It should be noted that the yield strength of the PC material had a linear relationship with the logarithm of the strain rate, which increased with the increase of the strain rate; the function of Equation (2) was used to fit, and the corresponding parameters a, b were obtained, where b corresponded to the yield strength of the PC material at a strain rate of 1/s. It can be seen that the yield strength of the two materials was basically the same, and the sensitivity to the strain rate was basically the same. In addition, for the A1225BK material, the yield strength was increased from 62.87 MPa to 81.68 MPa, and the yield strength was increased by 29.91%. Similarly, for the HL6157 material, the yield strength was increased from 62.51 MPa to 81.52 MPa, and the yield strength was increased by 30.41%. Overall, the A1225BK material and the HL6157 material increased the yield strength and tensile modulus when the strain rate increased, but the tensile modulus of A1225BK was more sensitive to the change of strain rate, and the sensitivity of the two materials in yield strength was roughly the same.

### 3.3. DIC Diagram

DIC technology is used to visualize the strain of materials at different rates, and explore pick points at different rates, as shown in Figure 6: ①the initial stage of elasticity, ② initial stage of yield (corresponding to yield strength), ③strain softening stage, ④plastic expansion stage, and ⑤ fracture stage.

Figure 7 illustrates the strain distribution diagram of the A1225BK material at two different strain rates, 0.01/s and 100/s, taking A1225BK as an example. At higher strain rates, the reflected macroscopic stiffness differed due to the inability of molecular segments of the PC material to move or rearrange in time, leading to a distinct strain distribution gradient. In the elastic stage, the strain distribution on the surface of the specimen was uniform at low strain rates (0.01/s), whereas a strain gradient appeared at high strain rates (100/s), indicating non-uniform strain distribution and inconsistent reaction times in different parts of the specimen during high-speed tension. However, after the end of the elastic region, the strain distribution pattern of the surface of the specimen at the two strain rates tended to be the same and was basically consistent, with a similar degree of necking observed in both. The initial position of the neck shrinkage mostly occurred in the middle of the specimen, suggesting that the primary difference between high-speed tension and low-speed tension lies in the elastic region. Following the onset of plasticity, the regular pattern for expansion of plasticity was substantially the same, indicating that the molecular chain movement and rearrangement in high-speed tension mainly occurred in the elastic region.

### 3.4. Microscopic Morphology of the Fracture

The fractures in the A1225BK and HL6157 specimens under different strain rates were scanned by electron microscopy to observe the differences in fracture morphology under different materials and different strain rates, and the mechanism of fracture was analyzed.

Figure 8 shows the tensile fracture morphology of the A1225BK material at different strain rates. Specifically, Figure 8(Ia) presents the overall fracture morphology of the specimen, while Figure 8(Ib–Id) shows the microscopic morphology of different regions. The fracture of the A1225BK material exhibited more residual resin and a rough overall texture after the static tension, as observed in Figure 8(Ia). The distribution of the radioactive stripes formed by the overall resin bulge on the surface of the fracture indicated that the source area of the fracture originated from the region shown in Figure 8(Ib), expanded into Figure 8(Ic), and finally led to the final fracture in Figure 8(Id). Comparing the microscopic morphology of the three regions, it can be seen that the protruding part of the specimen fracture in the Figure 8(Ia) region was the smoothest. In contrast, the final fracture area exhibited obvious tear-shaped remains of the PC material protrusion.

As shown in Figure 8II, an increase in the strain rate led to a smoother fracture of the specimen. The crack initiation was clearly observed in the region shown in Figure 8(IIb), followed by extension to the region shown in Figure 8(IIc), and ultimately resulted in fracture in the region shown in Figure 8(IId). The fan-shaped residual resin in the region shown in Figure 8(IId) indicated that the fracture occurred in this region. Moreover, the residual resin on the fracture surface in Figure 8(IIe) exhibited a more pronounced tear shape, confirming the ductile fracture behavior of the specimen at this location. These observations demonstrate that PC materials maintain their ductile fracture behavior even under high-speed tension and do not undergo brittle fracture. Therefore, the elongation at break of PC materials does not decrease after high-speed tension.

Compared with the results in Figure 8II, the fracture of the PC material did not show a further smooth trend when the strain rate was further increased, but also showed a more obvious surface bulge, which indicated that the PC material still showed obvious ductile fracture law at this strain rate. The protrusion at the fracture of the PC material started from the source area of the fracture, as shown in Figure 8(IIIb), and it was observed that the obvious crack or the source area of the hole-shaped nucleus was the surface of the specimen, which then extended to the entire fracture surface of the specimen; the smoothness of the fracture of the specimen also gradually increased with the direction of the stripe, but up to the final fracture area of the specimen, shown in Figure 8IIIe, the roughness of the fracture surface of the specimen increased, and the edge of the stripe showed obvious residual material caused by the ductile fracture.

The tensile fracture morphology of the HL6157 material under different strain rates is shown in Figure 9. Compared with the results of Figure 9I, it can be clearly observed that the fracture of HL6157 material was smoother and flatter, and the fracture did not show the PC material residue of obvious tear such as the A1225BK material. Similarly, in Figure 9(Ib–Id) of the HL6157 material, the fracture also behaved from the rough fault source region shown in Figure 9(Ib) to the smooth crack growth zone shown in Figure 9(Ic) and finally to the rough fracture zone shown in Figure 9(Id). Meanwhile, the source area of fracture was located in the corner of the specimen.

The fracture results of the two resins showed that under different strain rates, the fracture form of the A1225BK and HL6157 PC material was different, and the fracture of the PC material at a lower rate was mainly manifested as an obvious ductile fracture law. However, the ductile fracture degree was not reduced with the increase of the strain rate, but in the case of a higher strain rate, both resins obviously showed the law of ductile fracture, so the fracture effect of both resins showed good energy absorption at a high strain rate. It was beneficial to improve the impact resistance of the corresponding composite material. Meanwhile, the mechanical performance of A1225BK was more sensitive to the strain effect. Overall, the stiffness and fracture strain in A1225BK were bigger than those in HL6157 meaning A122BK is more suitable for improving the impact resistance.

## 4. Conclusions

In the case of high strain rate, the yield strength and modulus of PC material were significantly improved, and the modulus was proportional to the logarithm of strength and strain rate. At the same time, for the A1225BK and HL6157 materials, with the increase in strain rate, the fracture strain increase was greater than 10% and the yield strength increased by 30%.The microscopic characteristics of the fracture of PC material showed that the A1225BK and HL6157 materials still followed the law of ductile fracture at high strain rates, such as the resin residue at the break, which proves that they have good energy absorption ability and good fracture toughness at high strain rates.As the matrix for composite materials, A1225BK and HL6157 have good fracture toughness, which can be used to improve the overall impact resistance while meeting the lightweight design of automotives. Additionally, the higher stiffness and higher fracture strain make A1225BK more suitable for the design.

## Figures and Tables

**Figure 1 materials-16-03386-f001:**
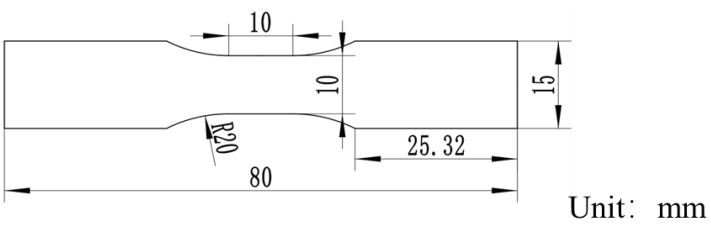
Geometry of the tensile specimen.

**Figure 2 materials-16-03386-f002:**
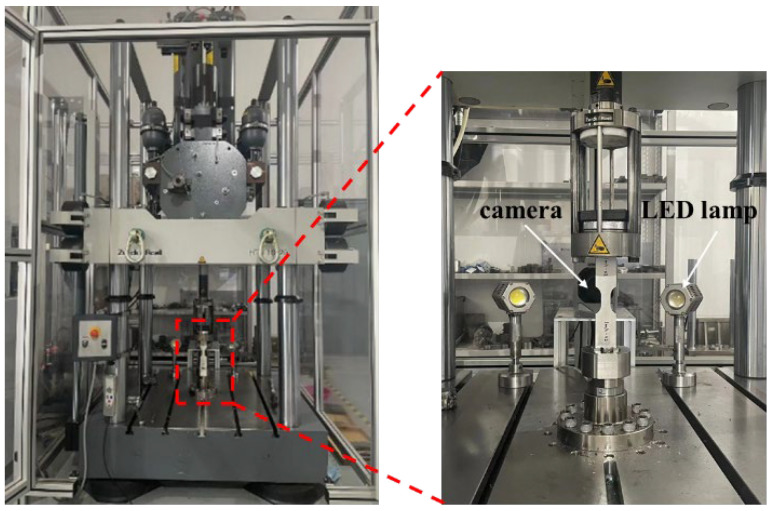
High-speed tensile test platform based on DIC technology.

**Figure 3 materials-16-03386-f003:**
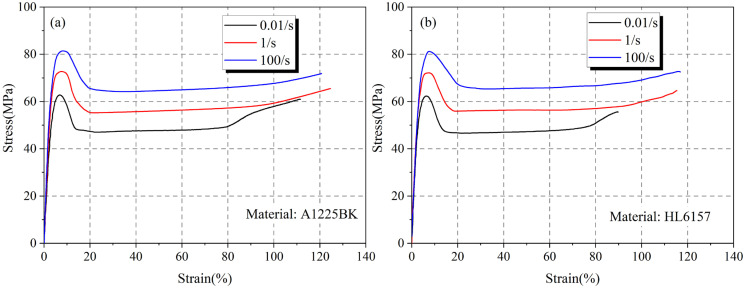
Stress–strain relationship curves of different materials at different strain rates (**a**) A1225BK (**b**) HL6157.

**Figure 4 materials-16-03386-f004:**
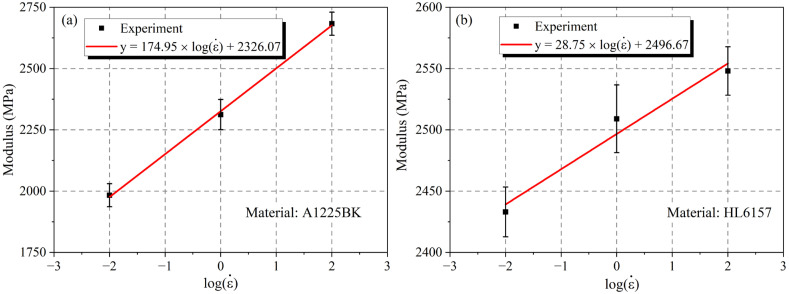
The relationship of the modulus of different PC materials and the logarithm of the strain rate (**a**) A1225BK (**b**) HL6157.

**Figure 5 materials-16-03386-f005:**
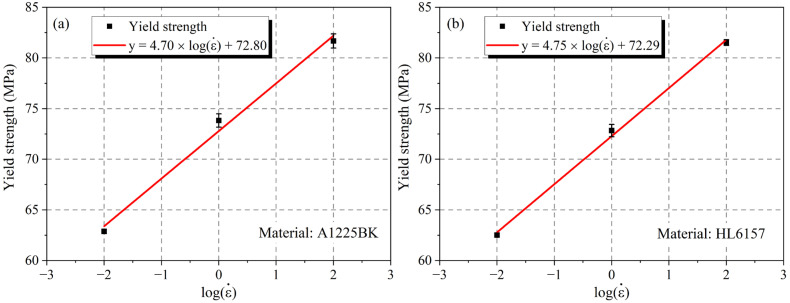
The relationship of the yield strength of different PC materials and logarithm of the strain rate (**a**) A1225BK (**b**) HL6157.

**Figure 6 materials-16-03386-f006:**
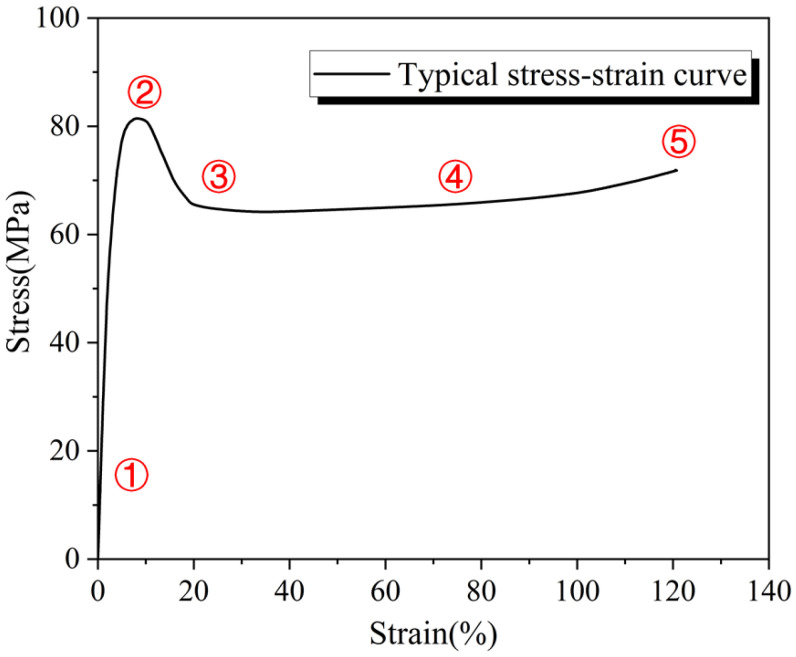
Schematic diagram of DIC selection point.

**Figure 7 materials-16-03386-f007:**
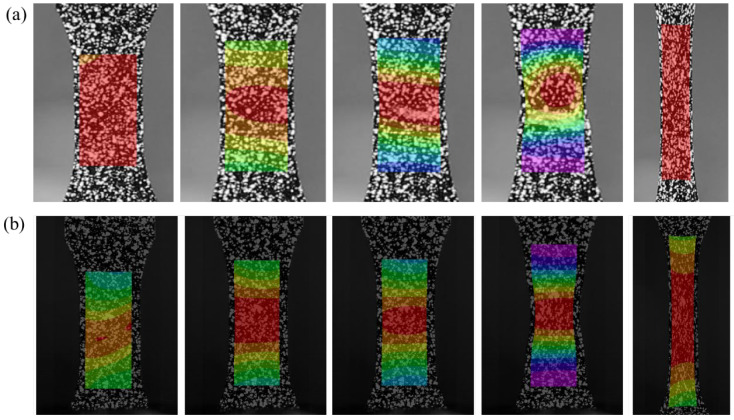
Strain distribution diagram of each node of A1225BK material (**a**) 0.01/s (**b**) 100/s.

**Figure 8 materials-16-03386-f008:**
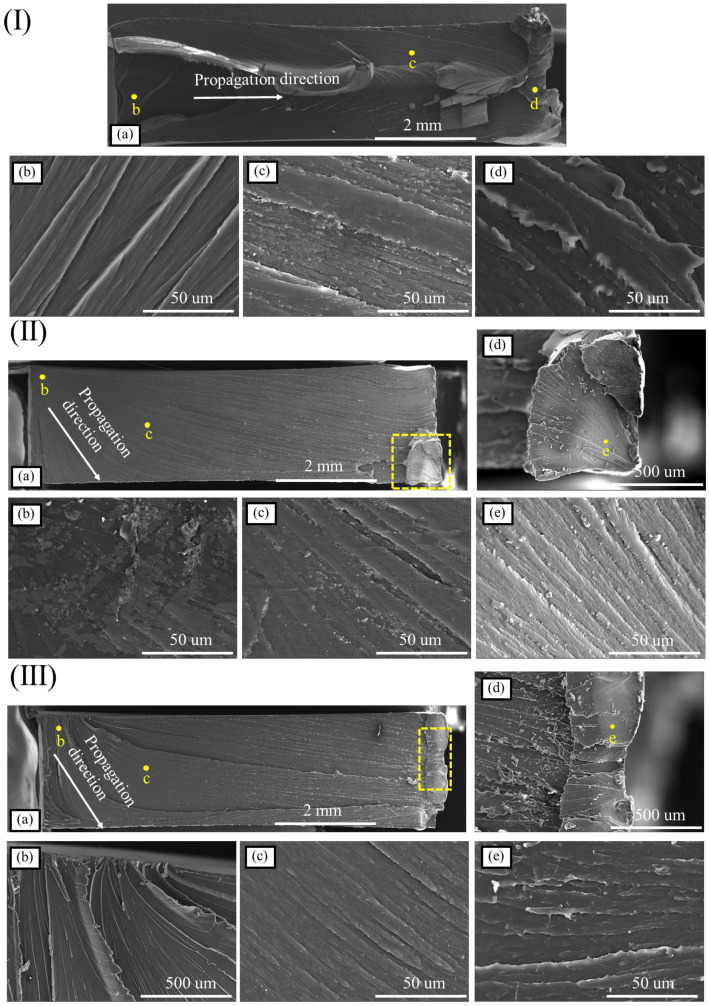
Fracture microstructure of the A1225BK material (**I**) 0.01/s—(**a**) Global diagram; (**b**) source area; (**c**) growth area; (**d**) fracture area; (**II**) 1/s—(**a**) Global diagram; (**b**) source area; (**c**) growth area; (**d**) fracture area; (**e**) local enlarged view of (**d**); (**III**) 100/s—(**a**) Global diagram; (**b**) source area; (**c**) growth area; (**d**) fracture area; (**e**) local enlarged view of (**d**).

**Figure 9 materials-16-03386-f009:**
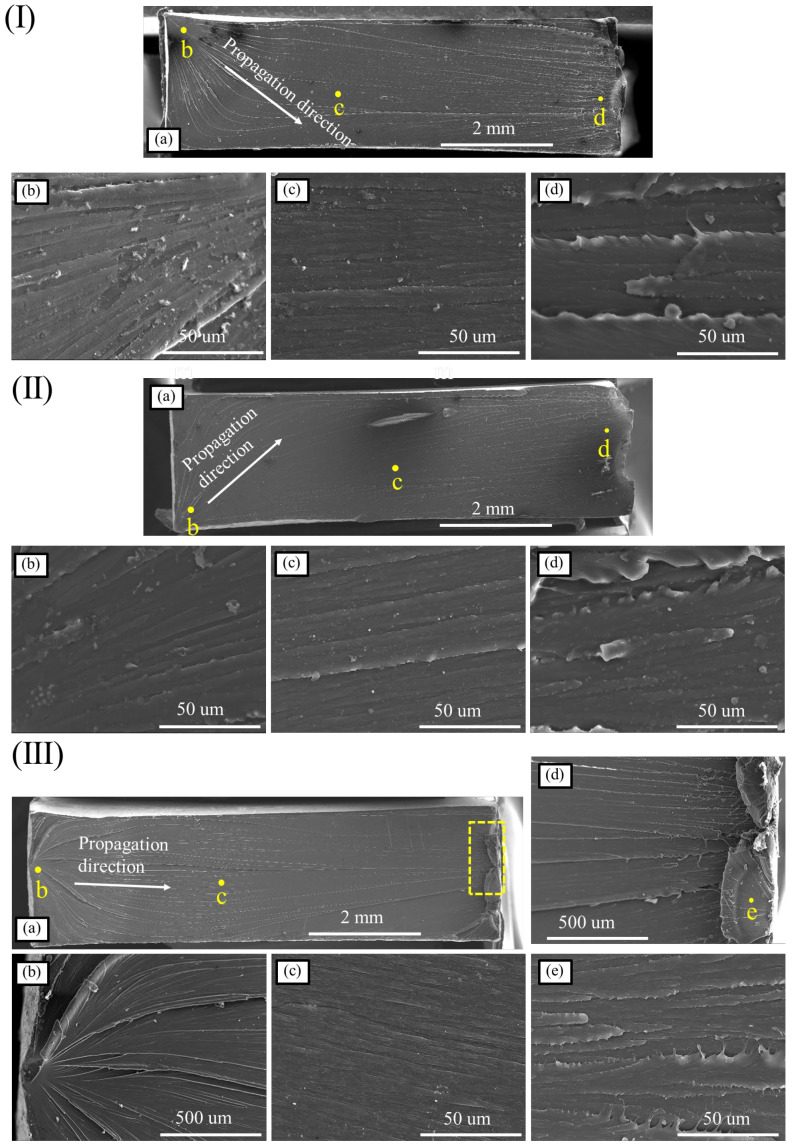
Fracture microstructure of the HL6157 material (**I**) 0.01/s—(**a**) Global diagram; (**b**) source area; (**c**) growth area; (**d**) fracture area; (**II**) 1/s—(**a**) Global diagram; (**b**) source area; (**c**) growth area; (**d**) fracture area; (**III**) 100/s—(**a**) Global diagram; (**b**) source area; (**c**) growth area; (**d**) fracture area; (**e**) local enlarged view of (**d**).

**Table 1 materials-16-03386-t001:** Process parameters of two materials.

Density (g/cm^3^)	Processing Temperature °C	Mold Temperature °C	Time h
1.18~1.20	240~250	50~80	8~10

## Data Availability

Not applicable.

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
