# Peer review of "Mechanical Properties and Fracture Microstructure of Polycarbonate under High Strain Rate Tension"

_materials, 2023, doi:10.3390/ma16093386_

Round 1
Reviewer 1 Report
The authors presented an article about “Mechanical properties and fracture microstructure of polycarbonate under high strain rate tension.” The authors examined the mechanical properties for two different polymer materials. It can be said that the layout of the material method section is particularly good in the paper. However, it can be said that the introduction part and the discussion part are quite short and simple. I think the paper is not well organized and appropriate for the “Materials” journal, but the paper will be ready for publication after major revision.
• The abstract looks good. Please include all significant numerical results.
• For the introduction section, please add more references (related to the article's topic (2018-2023)) and briefly explain them.
• What is the problem? Why was the manuscript written? Please explain the reason in the introduction part. In the last paragraph of the introduction, the novelty of the study and the differences from the past in detail should be expressed.
• Give information about the production of the materials used in the study.
• Are the mechanical tests carried out in accordance with the test standards? Please specify test standards in methods section.
• Figures 3-4-5-6 image quality is unprofessional. Please improve the image quality of these figures.
• Expand the discussion section based on similar studies in the literature. Please cite necessary literature research.
• Please fix the typographical and eventual language problems in the paper.
• The paper is well-organized, yet there is a reference problem. First, your reference list contains no paper from the “Materials” journal. If your work is convenient for this journal’s context, then there are many references from this journal. Secondly, cited sources should be primary ones. Namely, the indexed area shows the power of a paper and directly your paper’s reliability. Please make regulations in this direction.
*** Authors must consider them properly before submitting the revised manuscript. A point-by-point reply is required when the revised files are submitted.
Reviewer 2 Report
The manuscript is clear, relevant for the field and presented in a well-structured manner. The literature review should include references to more literature items. Cited references are generally recent publications (one literary item was issued many years ago). The work includes four self-citations of the first three authors of work.
The manuscript sound scientifically and was logically divided into chapters. The article concludes with brief and eligible conclusions.
Minor errors noticed by the reviewer are presented below:
· Line 32 – typo: “…carbon fiber is It is a brittle material”;
· line 37 – typo: ”… high strain rate Therefore, a high-speed tensile …”;
· line 69 – typo: “… heating panels, Frost and …”;
· line 119 - wrong caption under the figure 2;
· line 129 – typo: “…ultimately broken break In the area…”;
· line 132 – typo: “… (1/s and 100/s) The possible reason…”;
· line 135 – typo: ”… occurring suffered suffering…”;
· line 140 - typo: “… increases Highly increase…”;
· line 271 – typo: “…the which proves that…”;
· line 273-274 – repetition of the phrase: “… which can be used as a matrix for composite materials, which can be used…”.
A brief summary:
The paper is well written, well-structured and edited with care (small editing errors). The literature review contains few publications. The figures are nice. The article concludes with brief and eligible conclusions.
Reviewer 3 Report
The present manuscript is of average scientific interest. It may be accepted after author response to following comments.
1. Need to polish the language of the papers. Authors have repeated words such as “when” or “which” or “but” in the same sentence. So, my suggestion is to avoid such repetition. (for example, check lines 273- 275 and 258-260 and 130, 131.)
2. If possible, please improve the quality of figure 9. Since the words written in red are not clearly visible. Also check figure 8.
3. The author has not discussed about the Figure 8 II e as well as Figure 8III e in the main text.
4. Please separate the sentence “When the strain…………………………………………….fractured at this location.” (lines 226-230).
5. Please avoid the capital font of the first letter of words in the middle of sentences. Check the whole of the manuscript.
6. The author has discussed the importance of light weight composites for automobile application. In this context, it is my request to the author to give the density of the matrices employed.
7. I think captions for Figure 2 need to be written. Since author has written ‘This is a Figure’, Figure?
8. If possible please cite some of the latest works in lines 27& 28 (Composite materials …………. overall weight of the structure) as given below.
https://doi.org/10.1016/j.biortech.2022.128255
https://doi.org/10.3390/ma16030967
https://doi.org/10.1177/0731684412442989
9. Author in the abstract section has mentioned that yield strength of two PC materials increases by 30%. But in the results and discussion section, no such discussion has been made. If author has made a conclusion on the basis of figure 5, then please discuss it briefly in section 3.2.
1 10. At the end, author is requested to improve the quality of English.
Reviewer 4 Report
Manuscript of Sai Zhang et. al. is devoted to the search for a thermoplastic polymer matrix to obtain composites in order to facilitate structures and parts for transport. It is proposed to use polycarbonate (PC) as such a matrix. The purpose of this work is to study the mechanical behavior and peculiarities of destruction of two types of PC samples at three rates, including at a very high rate of 100 s-1. The main conclusion of the manuscript is that PC does not become brittle even at a very high tensile rate, retains its ability to plastic deformation, and its destruction has a ductile character. The paper itself is of a descriptive type, the results obtained are not discussed in any way, which reduces the significance of the work. There is no explanation for the differences in the mechanical behavior of the two types of PC, which differ in composition and additive content. The methodology does not specify how many specimens were tested for each tensile condition, nor does it provide a methodology for mathematical statistical processing of the values. Therefore, it is not clear how significant are the differences in the values of elongation at break and the modulus of elasticity (the error of the method is about 10%) obtained at different rates.
In my opinion, the paper as written has no scientific value and cannot be recommended for publication in Materials.
Author Response
Thanks for your attention. Five specimens are tested under each tensile condition, and make sure that five results are effective for each tensile condition, meanwhile, select the average as the final result.
Round 2
Reviewer 1 Report
Thank you for the revised manuscript
Author Response
Thank you for the review.
Reviewer 3 Report
Dear Author
I am satisfied with your response and thus recommending for its acceptance.
Author Response
Thank your for the review.